# Atomically dispersed MoNi alloy catalyst for partial oxidation of methane

Zheyuan Ding [1,2,6], Sai Chen[1,2,6], Tingting Yang[1,2,3], Zunrong Sheng[1,2], Xianhua Zhang[1,2], Chunlei Pei[1,2], Donglong Fu[1,2], Zhi-Jian Zhao [1,2] & Jinlong Gong [1,2,3,4,5] ✉

The catalytic partial oxidation of methane (POM) presents a promising technology for synthesizing syngas. However, it faces severe over-oxidation over catalyst surface. Attempts to modify metal surfaces by incorporating a secondary metal towards C–H bond activation of $CH_4$ with moderate O* adsorption have remained the subject of intense research yet challenging. Herein, we report that high catalytic performance for POM can be achieved by the regulation of O* occupation in the atomically dispersed (AD) MoNi alloy, with over 95% $CH_4$ conversion and 97% syngas selectivity at 800 °C. The combination of ex-situ/in-situ characterizations, kinetic analysis and DFT (density functional theory) calculations reveal that Mo-Ni dual sites in AD MoNi alloy afford the declined $O_2$ poisoning on Ni sites with rarely weaken $CH_4$ activation for partial oxidation pathway following the combustion reforming reaction (CRR) mechanism. These results underscore the effectiveness of $CH_4$ turnovers by the design of atomically dispersed alloys with tunable O* adsorption.

Partial oxidation of methane (POM) $(CH_4 + \frac{1}{2}O_2 \rightarrow CO + 2H_2$ $\Delta H°_{298K} = -35.5\,\text{kJ mol}^{-1})$ is considered as an energy-efficient process for syngas production, due to its exothermic nature and no need for adiabatic equipment[1–5]. In addition, a $H_2$:CO ratio of 2 is well suitable for downstream conversion such as methanol or Fischer–Tropsch synthesis[6–9]. Noble metal catalysts typically own excellent activity and selectivity at lower temperature, rare abundance and high price, however, limit its further large-scale application[2,7,10–17]. Supported Ni-based catalysts have been frequently used in POM reaction for its high activity and low cost with respect to noble catalysts[6,18].

In general, two main mechanisms, i.e., direct partial oxidation (DPO) and combustion reforming reaction (CRR), have been proposed as the main mechanisms schemed in Supplementary Fig. 1. The dissociated carbon species (C*) from $CH_4$ directly reacts with $O_2$ dissociated oxygen species (O*) in DPO mechanism while CRR mechanism involves first $CH_4$ combustion with $O_2$ and resultant $CO_2$ and $H_2O$ then reforming with $CH_4$ to generate syngas[19]. Accordingly, Ni supported on metal oxides (Ni/Oxides) has been designed to provide active oxygen species by incorporating rare earth elements such as Ce, La, Zr, Y, etc. La-Ce and Ce-Zr mixture oxides[5,13,20–23] were also used to enhance the properties of anti-coking and anti-sintering by active oxygen carriers and strong metal-support interaction[24,25]. However, the challenge of over-oxidation persists due to the fact that the same Ni site catalyses both the C–H activation and O addition, leading to a high propensity for over-oxidation[26–28].

The tuning of intermediate species C* and O* adsorption over metal surface is crucial for optimization of reaction pathway[29–35]. Previous reports indicated maintenance of moderate O* binding strength is conducive to the partial oxidation pathway under high reaction temperatures[5,13,20–23,36–40]. Alloying could provide constructive

[1]Key Laboratory for Green Chemical Technology of Ministry of Education, School of Chemical Engineering & Technology, Collaborative Innovation Center for Chemical Science & Engineering, Tianjin University, Tianjin 300072, China. [2]Collaborative Innovation Center for Chemical Science & Engineering (Tianjin), Tianjin 300072, China. [3]Joint School of National University of Singapore and Tianjin University, International Campus of Tianjin University, Binhai New City, Fuzhou 350207, China. [4]Haihe Laboratory of Sustainable Chemical Transformations, Tianjin, China. [5]National Industry-Education Platform of Energy Storage, Tianjin University, Tianjin, China. [6]These authors contributed equally: Zheyuan Ding, Sai Chen. ✉e-mail: jlgong@tju.edu.cn

geometric and electronic structures for optimum adsorption energy close to the volcano peak performance. For instance, the PtCu alloy was designed for propane combustion where Pt-Cu dual sites achieved a more favorable C* and O* binding strength than Pt[32]. The oxophilic nature of Mo owns stronger O* adsorption energy and electronegativity over Ni[41,42], therefore, the Mo-Ni dual sites might offer the facile C−H activation and O* adsorption towards partial oxidation of methane.

In this work, we employed a ligand protection method to design a catalyst based on atomically dispersed MoNi alloy (AD MoNi alloy) for the partial oxidation of methane. The well-defined electronic and geometric structure for AD MoNi alloy was evidenced by aberration-corrected high-angle annular dark-field scanning transmission electron microscope (AC-HAADF-STEM), X-ray absorption fine structure (XAFS), X-ray photoelectron spectroscopy (XPS) and X-ray radiation diffraction (XRD). It achieved high catalytic performance with over 95% $CH_4$ conversion and 97% syngas selectivity (800 °C). The partial oxidation of methane was evidenced following a CRR mechanism, as deduced from investigations into both steady and transient state temperature-programmed surface reaction kinetics. Furthermore, a detailed exploration from an atomic perspective illustrated that the atomically dispersed Mo sites played a crucial role in optimizing the adsorption of O* species. This optimization effectively mitigated O* over-occupation on Ni sites, thereby significantly promoting the partial oxidation pathway. These findings contribute to a comprehensive understanding of the mechanisms underlying the enhanced catalytic performance through the atomically dispersed MoNi alloy catalyst.

## Results
### Formation and structural characterization of atomically dispersed MoNi alloy

Bimetallic MoNi alloy nanocrystals were prepared with a ligand protection wet-chemistry method with various atomic ratios of Mo to Ni, as illustrated in Supplementary Fig. 2 (see details in Methods). It's noteworthy that employing excessive ligand protection strategies could provide effective protocols against oxidation or segregation during alloy preparation[42–50]. AD MoNi alloy and 5Ni1Mo alloy with Mo/Ni atomic ratio of ~1:20 and ~1:5 were evidenced by inductively coupled plasma optical emission spectroscopy (ICP-OES) and element dispersive spectroscopy (EDS) mapping shown in Supplementary Fig. 3. The AC-HAADF-STEM was further applied to provide the atomic resolution images of the Ni particles, AD MoNi alloy and 5Ni1Mo alloy, respectively (Fig. 1a–c). The bright dots, corresponding to Mo atom column, were contrasted with the darker Ni atom column (Fig. 1d–f). Atomic dispersion of Mo atom was observed for AD MoNi alloy (Fig. 1e) while multiple Mo atom with Mo-Mo doping exsisted in 5Ni1Mo alloy (Fig. 1f). This observation is consistent with the line profile along the lattice patterning in Fig. 1g–i. These alloys were located within face-centered cubic (FCC) solid solution alloy according to Mo-Ni phase diagram (Supplementary Fig. 4)[51,52]. The interplanar distance was determined to be 1.77 Å and 1.84 Å for AD MoNi alloy and 5Ni1Mo alloy, respectively, slightly larger than that of pure Ni (1.76 Å) towards <200> plane, also indicating the formation of MoNi alloy.

Further, local chemical environments of Ni including coordination and electronic structures were analyzed following the addition of Mo atoms using X-ray absorption fine structure (XAFS) spectra including X-ray absorption near edge structure (XANES) and extended X-ray absorption fine structure (EXAFS). The XANES spectra depicted in Fig. 2a could reflect the electronic structure variation between Ni and MoNi alloy, indicating electron transfer from Ni to Mo atoms. This observation aligns with the XPS results (Supplementary Fig. 5). In the EXAFS R-space spectra shown in Fig. 2b and Supplementary Fig. 6, only metallic Ni, characterized by the Ni-Ni feature, is evident with absence of Ni-O or Ni-Ni peaks associated with NiO. The Fourier transforms (FT) of $k^3$-weighted EXAFS R-space spectra proves Ni-Ni bond shifts to

longer radial distance from AD MoNi alloy at ~2.16 Å to 5Ni1Mo alloy at ~2.20 Å due to Mo atom doping. The diminishing peak intensity of the Ni-Ni bond suggests a damped coordination structure of Ni with an increasing Mo content. The EXAFS fitting of Ni K-edge, as illustrated in Supplementary Fig. 7 and summarized in Table S1, further reveals a corresponding decline in the Ni-Ni bond coordination number (CN) declining from Ni foil (12) to AD MoNi alloy (10.5) and 5Ni1Mo alloy (7.4)[42]. It proves Mo coordination with Ni may decrease CN of Ni-Ni. Wavelet transform (WT) was further performed towards Ni foil, AD MoNi alloy and 5Ni1Mo alloy in Supplementary Fig. 7, respectively, indicating single Ni-Ni peak feature for all samples. That means all samples exhibit metallic Ni features rather than nickel oxides. In summary, the combination of AC-HAADF-STEM, XAFS and XPS provides a comprehensive and solid proof for experimental identification of atomically dispersed MoNi alloy structure.

### Catalytic behaviour of AD MoNi alloy for POM

To test the catalytic performance of POM, pure Ni, AD MoNi alloy and 5Ni1Mo alloy were supported on SBA-15 (named as Ni/SBA-15, AD MoNi alloy/SBA-15 and 5Ni1Mo alloy/SBA-15) using impregnation method (see details in Methods). The initial methane conversion and syngas selectivity exhibit a volcano plot trend with increase of Mo doping ratios (Fig. 3a). Ni/SBA-15 as referring sample presents 97% initial methane conversion with relatively low syngas selectivity of ~90% at 800 °C. 5Ni1Mo/SBA-15 with higher Mo content shows lower activity with ~32% initial conversion and ~92% selectivity. By contrast, AD MoNi alloy/SBA-15 catalyst remarkably improves the performance with 95.2% initial methane conversion and a high syngas selectivity of over 97% (CO of 97.7% and $H_2$ of 98.3%) with a steady $H_2/CO$ ratio of ~2.0 at 800 °C (Fig. 3b, c). It is noted that the MoNi catalysts prepared by impregnation method exhibited much lower $CH_4$ conversion and syngas selectivity (Supplementary Fig. 8), might be due to the oxidation or segregation during alloy preparation[42–50]. The stability for AD MoNi alloy/SBA-15 was operated for 20 h, which proved that the alloy could maintain high selectivity towards partial oxidation path into syngas. The catalytic performance comparison is detailed in Table S3 and S4, highlighting the superior selectivity of the AD MoNi alloy/SBA-15 catalyst compared to previous reports. This catalyst demonstrates a remarkable selectivity of over 97%, higher than the most reported Ni-based catalysts and comparable to noble metal-based catalysts.

The catalytic performance under varying conditions, encompassing temperature and gaseous hourly space velocity (GHSV), for Ni/SBA-15 and AD MoNi alloy/SBA-15, is presented in Supplementary Fig. 9 and Supplementary Fig. 10. It appears that the AD MoNi alloy/SBA-15 maintains stability in the temperature range from 800 to 700 °C and under various GHSV ranging from 15,000 to 48,000 $mL_{CH4} g_{cat}^{-1} h^{-1}$. In contrast, the Ni/SBA-15 catalyst experiences a slight deactivation for methane conversion from 74.4% to 73.1% after 2 h reaction starting from 700 °C with a GHSV of 15,000 $mL_{CH4} g_{cat}^{-1} h^{-1}$. As the GHSV increases to 24,000 $mL_{CH4} g_{cat}^{-1} h^{-1}$ at the same temperature (800 °C), Ni/SBA-15 shows noticeable deactivation, reducing methane conversion from 85.5% to 80.3% after a 2 h reaction (Supplementary Fig. 9e), lower than that of 92.5% for AD MoNi alloy/SBA-15. Therefore, the comparison emphasizes the superior performance and stability of AD MoNi alloy/SBA-15 over Ni/SBA-15 under the investigated conditions shown in Tables S5–S7.

### Identification of active phases for POM

To elucidate the active phases, several control experiments were conducted using $MoO_3/SBA$-15, $NiO/SBA$-15 and pure SBA-15, all tested with similar loading under the same reaction conditions. Pure SBA-15 could rarely convert $CH_4$ with ~2% conversion and $MoO_3/SBA$-15 owns a 6.9% $CH_4$ conversion. Although $NiO/SBA$-15 shows an enhanced conversion of 16.8%, the byproduct $CO_2$ selectivity of 47% was substantial

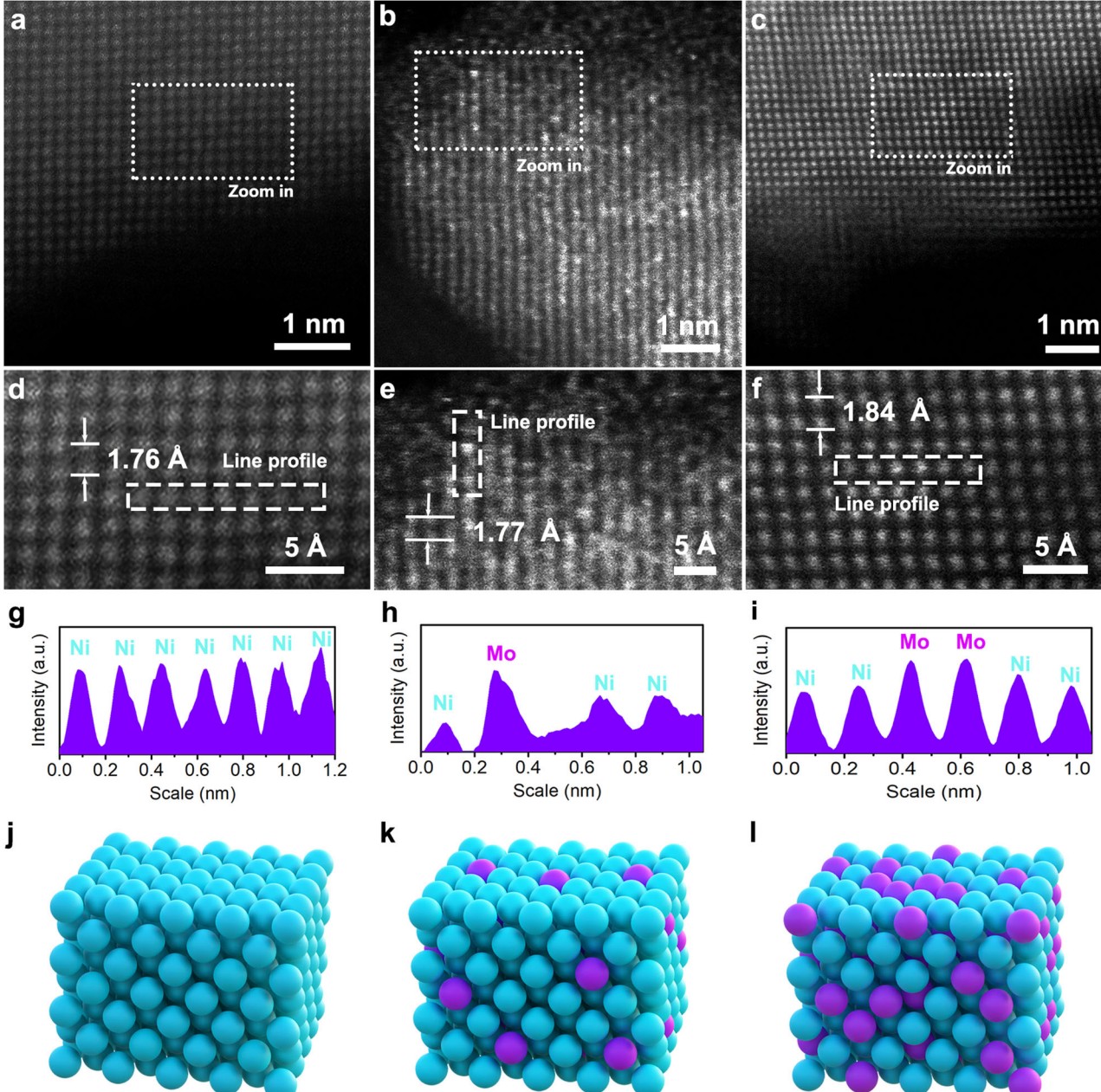

**Fig. 1 | Morphological characterization.** Representative aberration-corrected high-angle annular dark-field scanning transmission electron microscope (AC-HAADF-STEM) images of (**a**) Ni, (**b**) atomically dispersed MoNi alloy and (**c**) 5Ni1Mo alloy. **d–f** Zoom images for (**a**) Ni, (**b**) AD MoNi alloy and (**c**) 5Ni1Mo alloy, respectively. The bright dots corresponding to Mo atom in MoNi alloy. **g–i** Line profiles for (**d**) Ni, (**e**) AD MoNi alloy and (**f**) 5Ni1Mo alloy, respectively. The interplanar distances are determined along the lattice according to the arrow direction in (**d–f**). **j–l** Schemes of atomic structure for (**j**) Ni, (**k**) AD MoNi alloy and (**l**) 5Ni1Mo alloy, respectively. Ni: blue ball; Mo: purple ball.

(Fig. 3d). The control experiments suggest that the metallic Ni could serve as the active phase for POM rather than Mo oxide, NiO or SBA-15 support, which is consistent with previously common consciousness[19]. Additional analyses, including ex-situ XRD and EDS-mapping for spent Ni/SBA-15 catalyst shown in Supplementary Fig. 11, and major Ni phase transferred into NiO corresponding with the transformation of $Ni^0$ to $Ni^{2+}$ evidenced by quasi in-situ XPS after 1 h POM reaction at 800 °C, reflecting the formation of NiO phase is the main reason for the over-oxidation[53–55]. In-situ XRD (Supplementary Fig. 12a) demonstrated the stable metallic Ni phase over AD MoNi alloy/SBA-15 during time-on-stream testing condition at 800 °C with $CH_4/O_2/N_2$ ($CH_4:O_2 = 2:1$) flow. Therefore, it's anticipated that the MoNi alloy phase is intrinsically active and stable for POM.

## Exploration of the reaction mechanism

To shed light on the POM mechanism for AD MoNi alloy/SBA-15, steady and transient state temperature-programmed surface reaction of $CH_4$ and $O_2$ coupled with mass spectra (TPSR-MS) provided insightful observations. In Fig. 4a and Supplementary Fig. 13, combustion products of $CO_2$ and $H_2O$ formation were observed at low temperatures, while CO and $H_2$ of POM products were presented with increasing temperature. The critical observation is that only when $CO_2$ and $H_2O$ signals decrease, can an increase in CO and $H_2$ be observed, directly proving a combustion and $CH_4$ reforming process with $CO_2$ and $H_2O$. In turn, the potentially intermediated reactions were separately conducted to elucidate the combustion and reforming mechanism. The dry reforming of methane (DRM) and steam reforming of methane

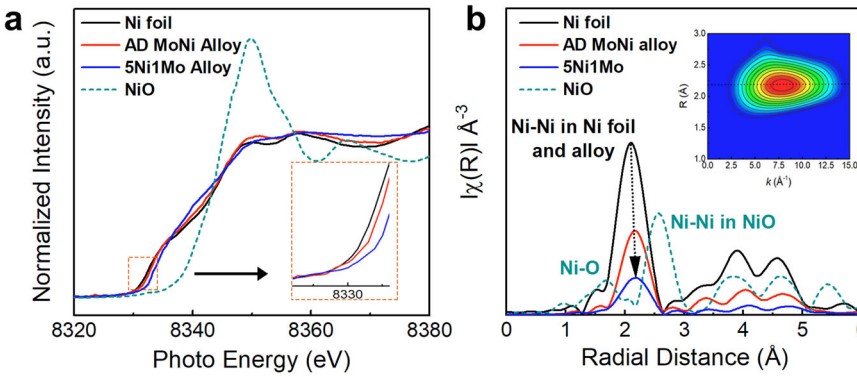

**Fig. 2 | Structural characterization. a** Ni K-edge XANES spectra for Ni foil, NiO, 5Ni1Mo and AD MoNi alloy, inset corresponding to enlarged zone indicative of increased oxidation state. **b** Fourier transforms (FT) of EXAFS R-space. Inset of (**b**)

corresponding to wavelet transform (WT) of AD MoNi alloy, indicative of single peak feature. 5Ni1Mo represents NiMo alloy with Ni:Mo ratio of 5:1 and AD MoNi alloy represents atomically dispersed MoNi alloy.

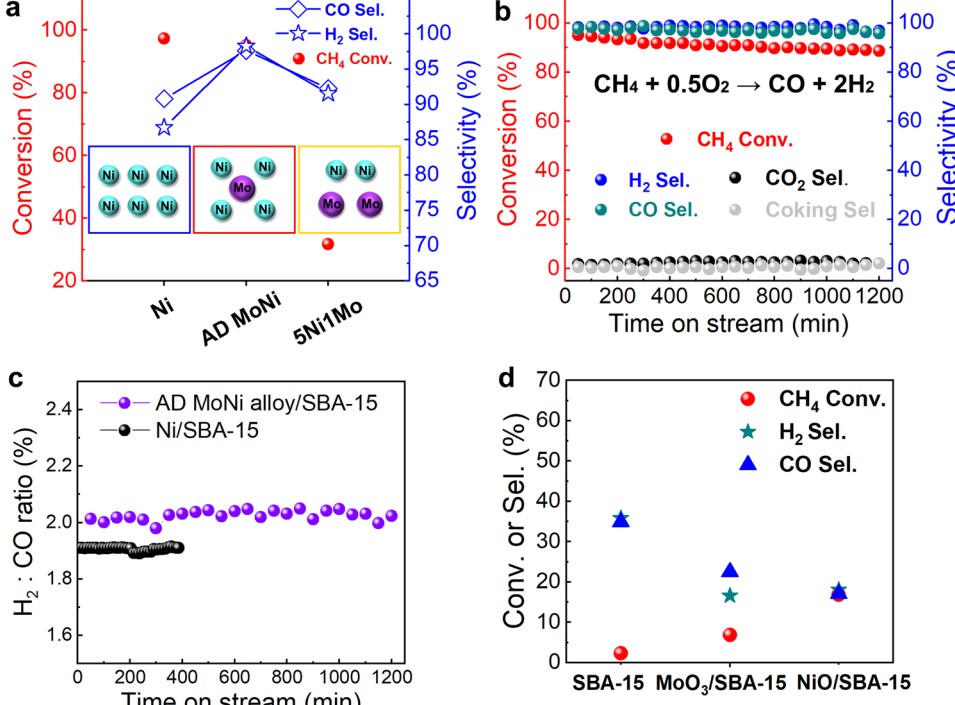

**Fig. 3 | Catalytic performance. a** Catalytic performance between Ni/SBA-15, 5Ni1Mo/SBA-15 and AD MoNi alloy/SBA-15 catalyst. Inset corresponding to schematic of representative structure. **b** Long-term stability test of AD MoNi alloy/SBA-15 under 800 °C in a $CH_4/O_2/N_2$ ($CH_4:O_2 = 2:1$) flow with GHSV of 12,000 $mL_{CH4}$ $g_{cat}^{-1}$ $h^{-1}$. **c** $H_2$: CO ratio of long-term stability test for AD MoNi alloy/SBA-15 and Ni/SBA-15

catalyst under 800 °C in a $CH_4/O_2/N_2$ ($CH_4:O_2 = 2:1$) flow with GHSV of 12,000 $mL_{CH4}$ $g_{cat}^{-1}$ $h^{-1}$. **d** Control experiments for SBA-15, $MoO_3$/SBA-15, NiO/SBA-15 and tested with similar loading weight under same condition. 5Ni1Mo represents NiMo alloy with Ni:Mo ratio of 5:1 and AD MoNi alloy represents atomically dispersed MoNi alloy.

(SRM) were then conducted for AD MoNi alloy/SBA-15 catalyst shown in Fig. 4b. DRM presents obvious reverse water gas shift (RWGS) reaction as $H_2 + CO_2 \rightarrow CO + H_2O$ with high CO selectivity of ~99% and ~84% $H_2$ selectivity while SRM exhibited a pronounced water gas shift (WGS) as $CO + H_2O \rightarrow H_2 + CO_2$ or coking with high $H_2$ selectivity of ~99% and ~81% CO selectivity. When the DRM and SRM (simultaneously adding $H_2O$ and $CO_2$ to reactor reforming with $CH_4$) is integrated, both CO and $H_2$ exhibit high selectivity ~98%, which could be attributed to the equilibrium of RWGS and WGS, contributing to a high selectivity for CO and $H_2$[56]. Overall, the TPSR-MS online experiments display a strong proof for POM mechanism consisting of combustion with reforming mechanism.

For the identification of C–H activation ability, $CH_4$ temperature-programmed surface reaction ($CH_4$-TPSR) tests were conducted for

both pure Ni/SBA-15 (Fig. 5a) and AD MoNi alloy/SBA-15 (Fig. 5b) catalysts. The close activation temperatures observed at 208 °C for Ni/SBA-15 and 216 °C for AD MoNi alloy/SBA-15 indicate that the addition of Mo for dilution of Ni did not significantly weaken C–H activation. The dispersion of Ni was determined by the $H_2$-pulse chemisorption experiments (Supplementary Fig. 14). The reaction order for $CH_4$ shown in Fig. 5c is near 1 for both Ni/SBA-15 and MoNi alloy/SBA-15, indicating a first-order reaction. This suggests that $CH_4$ activation serves as the rate-determining step (RDS). Reaction order for $O_2$ observed in Fig. 5d varied from 0.07 (Ni/SBA-15) to 0.31 (AD MoNi alloy/SBA-15), reflecting the strong $O_2$ adsorption and occupation on Ni/SBA-15 at 800 °C. Additionally, the turnover rates were also tested at a lower temperature of 670 °C (Supplementary Fig. 15) for both samples, and the reaction order for $O_2$ was shifted from −0.21

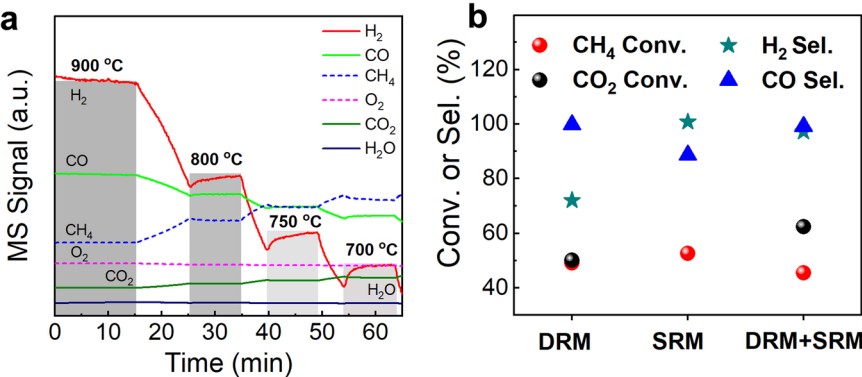

**Fig. 4 | POM mechanism identification. a** TPSR experiment for AD MoNi alloy/SBA-15 under testing condition with $CH_4:O_2 = 2:1$ flow for identification of POM mechanism. **b** DRM and SRM control experiments. DRM represents dry reforming of methane and SRM represents steam reforming of methane.

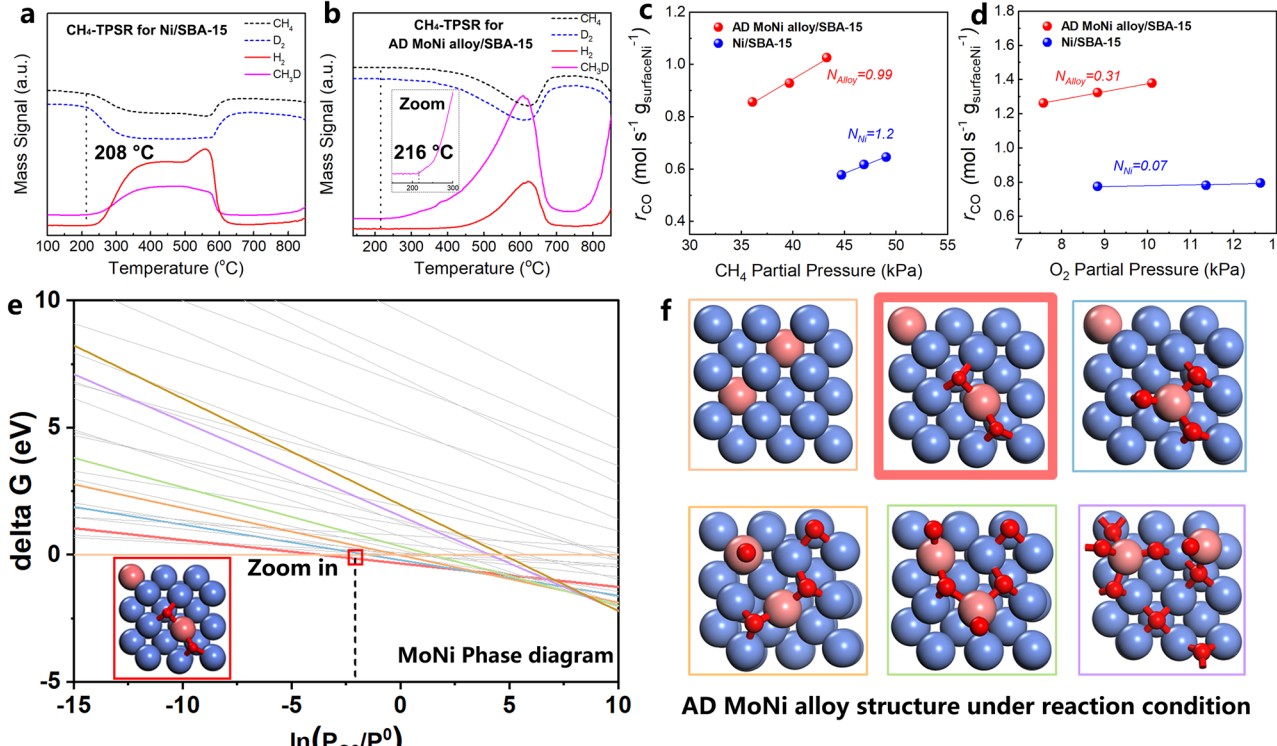

**Fig. 5 | Kinetic analysis and DFT calculation. a** $CH_4$-TPSR for pure Ni/SBA-15 and **(b)** AD MoNi alloy/SBA-15 catalyst. **c** Reaction order for $CH_4$ and **(d)** $O_2$. **e** Phase diagram for AD MoNi alloy with practical $O_2$ partial pressure of -12.5 kPa under reaction condition at 800 °C. **f** Possible structure of phase diagram in **(e)**. Red square in **(e)** and **(f)** represents real structure according to calculated phase diagram. Ni: blue ball; Mo: bright red ball; O: deep red ball. 5Ni1Mo represents NiMo alloy with Ni:Mo ratio of 5:1 and AD MoNi alloy represents atomically dispersed MoNi alloy.

(Ni/SBA-15) to 0.36 (AD MoNi alloy/SBA-15). This shift reflects competitive adsorption between $CH_4$ and $O_2$ following Langmuir-Hinshelwood model. $O_2$ poisoning on Ni sites is alleviated on the oxophilic Mo site in the alloy.

## DFT calculations

In addition, a density functional theory (DFT) study was conducted to provide a molecular-level insight into the mechanism of the partial oxidation of methane (POM) process. Energy calculations for a series of slab models were performed to determine the position of Mo in the Ni lattice. It was observed that Mo atoms located at the subsurface exhibited the most negative energy, as depicted in Supplementary Fig. 16. This suggests a preference for Mo atoms to reside within the subsurface layer of the MoNi (100). However, under reaction

conditions, the presence of surface-adsorbed oxygen species may induce the segregation of Mo atoms from the subsurface to the surface. Calculations of the segregation energy ($E_{seg}$) and oxygen Gibbs free adsorption energy ($E_{O-ads}$) revealed that as the number of surface $O^*$ species increased, the $E_{seg}$ of Mo became more negative and $E_{O-ads}$ is negative for 2–3 O atoms (Supplementary Fig. 17 and Supplementary Fig. 18). This indicates that Mo is more likely to migrate to the surface and occupy surface positions and only when 2-3 O atoms coverage could MoNi (100) remain stable. Subsequently, phase diagrams for MoNi (100) and Ni (100) were computed to elucidate the authentic surface structures of the catalysts under reaction conditions (Fig. 5e and Supplementary Fig. 19). Specifically, it was observed that there are no adsorbed oxygen species on the Ni (100) surface, while two adsorbed oxygen species were present on the MoNi (100) surface. To

confirm the stability of the MoNi structure with two pre-adsorbed O species, three repeated ab initio molecular dynamics (AIMD) simulations were conducted (Supplementary Fig. 20). The results of these simulations support the stability of the MoNi structure under the specified conditions.

We conducted a calculation of the reaction pathway for methane dehydrogenation on Ni (100) and MoNi (100). Schematic structures of all transition states are provided in Supplementary Fig. 21 and Supplementary Fig. 22. The activation energy barrier for methane on MoNi (100) is slightly higher than that on Ni (100), measuring 1.12 eV and 1.09 eV, respectively. This suggests a modest weakening of methane activation ability in the MoNi alloy, in line with experimental findings (Fig. 5). Subsequently, $CH_x^*$ species undergo dehydrogenation on Ni sites until forming $C^*$, which preferentially reacts with $O_2$ to form $CO_2$. Then, $2H^*$ forms $H_2O$ by interacting with pre-adsorbed $O^*$ on the MoNi surface. For methane combustion, the rate-determining step for both MoNi (100) and Ni (100) is the migration of $H^*$ from Ni to $O^*$, forming $HO^*$, with similar energy barriers observed for both surfaces[57]. However, the elementary step of $C^*$ combining with $O^*$ to form $CO^*$ is identified as the rate-determining step in the subsequent methane reforming reaction[57,58], as shown in Supplementary Fig. 23.

On MoNi (100), the adsorbed $O^*$ on Mo first migrates to Ni before forming adsorbed $CO^*$, rather than directly combining with C to form $CO^*$. Additionally, the energy barrier of $CO^*$ formation on MoNi (100) is observed to be lower than that on the Ni (100) surface, indicating that MoNi promotes the formation of CO, thereby enhancing product selectivity. To further explore the effects of Mo doping, we investigated $O_2$ adsorption behavior on MoNi (100) and Ni (100). Our findings reveal direct dissociation of $O_2$ on both surfaces, with significantly lower dissociation energy on pure MoNi compared to Ni and MoNi-2O surfaces (Supplementary Fig. 23c). This implies rapid replenishment of pre-adsorbed $O^*$ species on MoNi after consumption during the reaction, facilitating reaction progress. Thus, Mo primarily promotes the activation of $O_2$, inhibits the dissociative adsorption of $O_2$ on Ni sites on MoNi, and favors the enhancement of syngas selectivity, consistent with experimental results.

To summarize, we successfully synthesized atomically dispersed MoNi alloy catalysts through a refined ligand protection wet-chemistry method. The partial oxidation of methane reaction was elucidated to follow a combustion reforming reaction (CRR) mechanism based on TPSR and temperature distribution experiment. Kinetic analysis and DFT calculations showed that the introduction of oxophilic Mo adding is advantageous for activating $O^*$ species, alleviating the occupation of $O^*$ on Ni site open for methane activation. This design strategy resulted in a highly selective partial oxidation pathway with over 95% $CH_4$ conversion and over 97% syngas selectivity (800 °C). The modulation of $O^*$ adsorption strength in the atomically dispersed MoNi alloy pave a bright way for efficient methane high-valued conversion, offering promising prospects for selective oxidation and upgrading of light alkane in heterogeneous catalysis.

## Methods
### Catalyst preparation
The MoNi alloy nanocrystals (NCs) with varying MoNi ratios were prepared by a wet chemistry alloying approach. In a three-neck flask, a volume of solution corresponding to 1 mmol of Nickel (II) acetylacetonate (Ni(acac)$_2$, 95%, Sigma-Aldrich) was combined with variable quantities of Molybdenum (0) hexacarbonyl (Mo(CO)$_6$, 98%, Sigma-Aldrich) to achieve the desired MoNi ratio (1:20, 1:5 and 0). Oleylamine (OLAM, 10 mL) and excessive 1,2-Dodecanediol (~4 g) was in turn added to the bottle, and the mixture was placed under magnetic stirring and evacuated at 60 °C for 20 min. The reaction vessel was then switched into nitrogen flow as protecting gas during the reaction and heated quickly (~20 °C min$^{-1}$) to 230 °C. After 20 min of reaction, the solution color changed from cyan to brown finally to the

black and it was cooled rapidly to room temperature. The NCs were isolated by three rounds of precipitation using ethanol as anti-solvent with four times washing by hexane to remove excessive surfactant, and separated by centrifugation (8000 rpm, 5 min) in ~5 g solution. Finally, the NCs were dispersed in hexanes (~5 mL) for colloidal stability.

Pure Ni, AD MoNi alloy and 5Ni1Mo alloy were supported on SBA-15 mesoporous silica sieve (named as Ni/SBA-15, AD MoNi alloy/SBA-15 and 5Ni1Mo alloy/SBA-15). The as-synthesized NCs were rapidly impregnated onto SBA-15 supports (JCNANO Com.) with ultrasonic treatment (10 min) for well dispersion. After overnight drying at 70 °C, the powder was firstly reduced at 550 °C for 2 h and then raised to 800 °C for 2 h under flowing 10% $H_2/N_2$ to eliminate the residual surfactant and solvent for further use. The pure Ni/SBA-15 as reference used Ni(NO$_3$)$_2$ as Ni precursor and all ramping procedure and gas was similar. The MoO$_3$/SBA-15 was prepared by dissolving (NH$_4$)$_2$MoO$_4$ solution mixed with SBA-15 support and mixture was placed under magnetic stirring and evacuated at 60 °C. After overnight drying, it was then calcined at 800 °C for 2 h. The NiO/SBA-15 precursor was replaced as Ni(NO$_3$)$_2$ and all procedure was similar to MoO$_3$/SBA-15. Pure SBA-15 was untreated.

### Catalytic tests
Catalytic performance tests were performed in a quartz fixed-bed reactor with 8 mm inner diameter and 60 cm length at atmosphere pressure in furnace (BTF1200C, Anhui BEQ Equipment Technology Co., Ltd). 50 mg of catalyst with particle size of 20 to 40 meshes mixed with inert quartz sand was packed inside the quartz tubular reactor. The sample was firstly heated to 800 °C at a rate of 15 °C min$^{-1}$ and retained at 800 °C for 1 h in flowing 10% $H_2/N_2$, then switch to pure $N_2$ to purge the remaining $H_2$ before every test. During the catalytic activity test, a mixture of $CH_4$ (Air Liquide, 99.9%) and Air (Air Liquide) was fed at a rate of 35 mL min$^{-1}$ with a $CH_4$: $O_2$ ratio of 2:1 over 0.05 g catalysts. The gas hourly velocity (GHSV) of methane was about 12,000 mL$_{CH4}$ g$_{cat}$$^{-1}$ h$^{-1}$. The gas products were analyzed by an online GC (2060) equipped with thermal conductivity detectors (TCD) and TDX-01 column followed by a 5 A molecular sieve column.

The product conversion (Con.) and selectivity (Sel.) was calculated by the Eqs. (1) and (2) according to stoichiometric number:

$$Con.\,(\%) = 100 \times \left( [F_{CH4}]_{inlet} - [F_{CH4}]_{outlet} \right) / [F_{CH4}]_{inlet} \tag{1}$$

$$\begin{aligned} H2\,Sel.\,(\%) &= 100 \times [F_{H2}]_{outlet} / 2 \times \left( [F_{CH4}]_{inlet} - [F_{CH4}]_{outlet} \right), CO\,Sel.\,(\%) \\ &= 100 \times [F_{CO}]_{outlet} / \left( [F_{CH4}]_{inlet} - [F_{CH4}]_{outlet} \right) \end{aligned} \tag{2}$$

### Microscopy techniques
The samples were dispersed in ethanol or deionized water by ultrasonic for good dispersion and deposited on a Cu grid coated with ultrathin holey carbon film. Transmission electron microscopy (TEM) and energy-dispersive X-ray spectroscopy (EDS) mapping images were recorded on a JEOL JEM-F200 field emission transmission electron microscope operating at 200 kV. The aberration-corrected high-angle annular dark-field scanning transmission electron microscopy (AC-HAADF-STEM) was performed on a JEM-ARM200F operating at 200 kV capable of sub-angstrom resolution imaging.

The AC-HAADF-STEM $Z$-contrast imaging strongly depends on the sample atomic number $Z$ and is proportional to $Z^2$ reflected in Rutherford scattering cross-section. The $Z$-contrast imaging equation was noted as Eqs. (3) and (4).

$$\sigma = \left( \frac{m}{m_0} \right) \frac{Z^2 \lambda^4}{4\pi^3 \alpha_0^2} \left( \frac{1}{\theta_1^2 + \theta_0^2} - \frac{1}{\theta_2^2 + \theta_0^2} \right) \tag{3}$$

$$I_s = \sigma N t I \tag{4}$$

The $\sigma$ term reflect the Rutherford scattering cross-section over scatting angle from $\theta_1$ to $\theta_2$ and $\theta_O$ is Boone characteristic scattering angle. $Z$ is atomic number (Ni: 28, Mo: 42), $\alpha_O$ corresponded to Bohr radius, and $\lambda$ corresponding to wavelength, $m$ and $m_O$ is the electron mass of high-speed electronics and stationary electron. $I_s$ was denoted to scatting intensity, where $N$ is number of atom volume per unit among the thickness of $t$ to the sample and $I$ represent single atomic column scatting intensity.

## XAS

XAS experiments were conducted on 1W1B-XAFS at Beijing Synchrotron Radiation Facility (BSRF) with photo energy of 4-23 keV for Ni K-edge (8333 eV) and Mo K-edge (20000 eV) XAFS spectra. The storage ring was working at the energy of 2.5 GeV with 200 mA. A double crystal monochromator Si (111) was used with monochromatic X-ray incident beam at Ni K-edge. Data were acquired in transmission/fluorescence mode using Lytle detector. Spectra were acquired under room temperature and the edge jump should meet the requirement for >0.3. For energy calibration, reference Ni foil was firstly tested before every samples testing. The highest peak of 1st derivative of X-ray absorption near edge spectroscopy (XANES) was defined to be adsorption energy ($E_O$) for Ni K-edge $E_O$ calibration (8333 eV).

XAS data were processed and analyzed using the Demeter 0.9.26 software package fitted with Artemis and Athena software. The wavelet transformation was performed by HAMA Fortran package. The $k^3$-weighted EXAFS spectra were obtained by subtracting pre-edge and post-edge backgrounds from the overall absorption and then the edge jump was normalized using Athena software. The R space was obtained with fourier transforms of $k^3$-weighted $\chi(k)$ data with a Hanning window over a k range of 3.0-12.3 Å$^{-1}$ and fit in R space from 1.0-3.0 Å to separate the signal contributions from different coordination shells. The $k^3$-weighted $\chi(k)$ data were Fourier transformed after applying a IFEFF 1.2.12. The E-K transfer equation was factored with Eq. (5). The global amplitude EXAFS equation was noted as Eq. (6).

$$k = \frac{\sqrt{2m(E - E_0)}}{h} \tag{5}$$

$$\chi(k) = \sum_{i=1}^{shell} \frac{N_i S_0^2}{k R_i^2} F_i(k) e^{-2k^2 \sigma_i^2 k} \, e^{-\frac{2R_i}{\lambda(k)}} \sin\left[2kR_i + \varphi_i(k)\right] \tag{6}$$

Among the EXAFS fitting, the following factors were considered as the constant, variables, and evaluating standard for the goodness of fitting. $N_i$: coordination numbers; $R_i$: bond distance; $\sigma^2$: Debye-Waller factors; $\Delta E_O$: the inner potential correction. $R factor$: goodness of fit. $S_0^2$ for Ni-Ni was set as 0.98, which was obtained from the experimental EXAFS fit of NiO reference by fixing $N$ as the known crystallographic value and was fixed to all samples. In specific, $R$, $N$, and $\sigma$ were floating variables, while $S_0^2$ was fixed as 0.98 from a fitting to a NiO reference.

The neighboring atoms are divided into $i$ shells with coordination numbers $N_i$ of backscattering atom at a bond distance $R_i$ of absorber-scatter (e.g., metal-ligand or metal-metal) distance from the central atom. $N_i$ will linearly increase the amplitude of EXAFS signal, and $R_i$ contribute to EXAFS with $1/R^2$ dependence enabling EXAFS a local technique. The $F_i(k)$ term denoted to scattering amplitude function for each shell. $S_0^2$ is the amplitude reduction factor and is intrinsically related to the central atom, originating from shakeup process at central atom[59,60]. $S_0^2$ was set to be 0.98 in and kept constant in the EXAFS fitting of MoNi Alloy NCs with different atomic ratio. The Debye–Waller factor term, $e^{-2k^2 \sigma_i^2 k}$, reflected the disorder degree due to thermal or structural disorder. Amplitude losses from the inelastic scattering are factored with the mean free path term, $e^{-2R_i/\lambda(k)}$, where

$\lambda(k)$ corresponding to photoelectron mean free path. The periodicity in X-ray absorption coefficient could be reflected in sinusoidal term, $\sin[2kR_i + \varphi_i(k)]$, where $\varphi_i(k)$ equaled to phase shift function.

## ICP, XRD and ex-situ XPS analysis

Elemental composition of the catalysts was analysed by inductively coupled plasma optical emission spectroscopy (ICP-OES) (VISTA-MPX, Varian). Before measurements, the catalysts were digested in the mixed solutions of $HNO_3$ and HCl (1:3).

X-ray diffraction (XRD) patterns were obtained with a Bruker D8 using Cu K$\alpha$ radiation with wavelength $\lambda$ of 1.5418 Å operating at 40 kV, 200 mA. The in-situ XRD pattern was recorded on a Rigaku Smartlab diffractometer operating at 40 kV and 200 mA, and the graphite filtered Cu K$\alpha$ with wavelength $\lambda$ of 1.5418 Å was employed as the radiation source. The data points were collected by step scanning with a rate of 10° min$^{-1}$ from 20° to 60° (2$\theta$). X-ray photoelectron spectroscopy (XPS) experiment was performed on a Thermo Scientific K-Alpha$^+$ photoelectron spectrometer using Al K$\alpha$ (1486.6 eV) radiation.

## Quasi in-situ XPS

Quasi in-situ X-ray photoelectron spectroscopy (XPS) was performed using Al K$\alpha$ X-ray radiation (1486.6 eV) on a Thermo Fisher ESCALAB-Xi for spectra. The C 1s peak at 284.8 eV for binding energy (BE) correction was chosen as reference. The treatment and transfer of samples was totally performed in a ultra high vacuum (UHV)-connected high-pressure chamber. Samples were pre-treated in 10% $H_2$/Ar flow under ambient pressure at 600 °C for 60 min for every test. After in-situ POM reaction under reaction gas of $CH_4/O_2$/Ar (3:1.5:95.5) with total flow of 50 $sccm$ at 800 °C and purge by Ar stream for 1 h then cooled down to room temperature. The samples were transfered into the analyzer chamber totally under UHV environment during every transfer process and finally the photoelectron spectra were collected.

## TPR

The $CH_4$ temperature-programmed surface reaction ($CH_4$-TPSR), $CH_4$-$O_2$ temperature-programmed surface reaction ($CH_4$-$O_2$-TPSR) tests was performed on chemisorption facility (Micromeritics AutoChem II 2920) connected with mass spectrometer (MS) for the detection of product signal instantaneously. For $CH_4$-$D_2$-TPSR and TPSR experiments, 50 mg of sample was first reduced in flowing 10 vol% $H_2$/Ar at 600 °C for 1 h. Then, the sample was cooled down to 50 °C and the gas was flushed with flowing Ar to purge the catalyst. Subsequently, the $CH_4$ or $CH_4$-$O_2$ mixed gas was introduced into the apparatus with temperature raised at a rate of 10 °C/min up to 900 °C for $CH_4$-$D_2$-TPSR MS and $CH_4$-$O_2$-TPSR MS pattern, respectively. The MS signal of $CH_4$, $D_2$, HD, $CH_3D$ (m/z of 16, 4, 3 and 17) were recorded for $CH_4$-$D_2$-TPSR, and $CH_4$, $O_2$, $H_2$, CO, $CO_2$ and $H_2O$ (m/z of 16, 32, 2, 28, 44 and 18) were recorded for $CH_4$-$O_2$-TPSR. After that, the sample was purged with flowing He and cooled down to 50 °C.

## Dispersion

$H_2$-pulse chemisorption tests were conducted to measure the dispersion of Ni on each catalyst towards 50 mg of samples. Prior every measurement, catalysts were reduced at 650 °C for 1 h with a 10 vol. % $H_2$/Ar gas flow, and then cooled down to 50 °C under continuing Ar stream purge. After the TCD baseline was stable, $H_2$ was pulsed over catalyst until the consumption peaks area of $H_2$ remained stable. The dispersions were calculated based on the equation: Dispersion (%) = $100 \times V_{H2} \times 1/2 \times MW_{Ni} / (W_{Ni} \times 22414)$. where $V_{H2}$ is the volume of adsorbed $H_2$ (mL), $MW_{Ni}$ is the atomic weight of Ni (g·mol$^{-1}$), and $W_{Ni}$ is the weight of Ni supported on the sample (g), stoichiometry factor of Ni / $H_2$ equal to 2/1. The results were shown in the Supplementary Fig. 14. Dispersion of Ni sites in Ni/SBA−15 and AD MoNi alloy/SBA−15 catalysts were calculated as 1.7% and 0.93%, respectively.

## Kinetics tests

The kinetic experiments of POM on pure Ni/SBA-15 and AD MoNi alloy/SBA-15 catalysts were conducted in the fixed bed reactor, 1.2 mg Ni/SBA-15 or 0.1 mg AD MoNi alloy/SBA-15 mixed with quartz sand with total volume of 1 mL was loaded into a quartz tube, respectively. The kinetics study for AD MoNi alloy/SBA-15 catalyst was performed at temperature of 800 °C, while kinetic process of pure Ni/SBA-15 was performed in the temperature range of 670 to 800 °C. All the kinetics studies of POM were performed with conversions of CH₄ lower than 10% to ensure all data is tested under kinetics regime and assure there is no internal and external diffusion influences. The test condition was conducted under $CH_4$: $O_2$: $N_2$ flow of 2:1:5 over 0.1 mg AD MoNi alloy/SBA-15 catalyst (total flow rate 80 mL min$^{-1}$) and 1.2 mg Ni/SBA-15 (total flow rate 160 mL min$^{-1}$). The conversion and selectivity is about 2% and 96% for AD MoNi alloy and about 7.8% and 80% for Ni/SBA-15, respectively.

Rate correction process is conducted as follows. Considering the first-order deactivation kinetics, $r_t = r_{i,O}*e^{\wedge}(-k_i*t)$ where $r_t$ is forward rate at time $t$ and $k_i$ ($i = 1,2,3,4$) is the deactivation rate constant. The $r'_t$ is real forward rate at beginning time for every pressure condition. The term $r'_{i,O}$ is the corrected forward rate without deactivation at zero time and could be obtained by $r'_{i,O} = r_{i,O}*c_{i-1}$ ($i = 2,3,4$) where $r_{i,O}$ is real forward rate at zero time for every pressure condition. The deactivation coefficient $c_i$ ($i = 2,3,4$) can be obtained from $c_i = r_{i,O}/r_{i,ti}$ ($i = 1,2,3,4$) and $r_{i,ti}$ corresponding to end time $t_i$ ($i = 1,2,3,4$) of each pressure condition. The $r'_{i,O}$ is calculated and shown in the run log for AD MoNi alloy/SBA-15 and pure Ni/SBA−15 could be observed in Supplementary Fig. 15a and S15b. The correction results for Ni could be noticed in Supplementary Fig. 15c.

## Computational details

The density functional theory (DFT) calculations were performed with the Vienna ab initio simulation package (VASP)[61] using the generalized gradient approximation[62] in the form of the BEEF-vdw corrections[63]. The projector-augmented wave (PAW) method[64] with cutoff energy of 400 eV was used. Energies and forces for geometric optimization were converged to within 10$^{-5}$ eV and 0.02 eV/A, respectively. The MoNi (100) and Ni (100) were modeled by a five-layer slab 3 × 3 supercell, with the bottom two layers fixed. A 3 × 3 × 1 Monkhorst-Pack k-points grid was chosen to give an accurate sampling of the Brillouin zone. A 15 Å vacuum along the z direction was added to separate from the periodically repeated images. The transition states were searched by the climbing-image nudged elastic band (CI-NEB) method[65].

The activation energies of methane on MoNi (100) and Ni (100) were calculated as follows Eq. (7)

$$E_a = E_{TS} - E_{IS} \qquad (7)$$

where $E_{IS}$ and $E_{TS}$ are the energies of the initial state (IS) and transition state (TS), respectively.

The segregation energy was calculated by the following Eq. (8):

$$E_{seg} = E_{surf} - E_{sub} \qquad (8)$$

where $E_{surf}$ and $E_{sub}$ are energies of MoNi after and before Mo segregation from subsurface to surface.

Thermodynamic surface phase diagrams were calculated to determine the surface structure of Ni(100) and NiMo(100) under reaction condition. The surface energies are obtained as following Eq. (9):

$$\Delta G(T,p,N_O) = G_{slab}(T,p,N_O) - G_{clean}(T,p,N_O=O) - \frac{N_O}{2}\mu_{O_2}(T,p) \qquad (9)$$

where $G_{slab}(T,p,N_O)$ and $G_{clean}(T,p,N_O=O)$ are Gibbs free energies of surfaces (Ni or MoNi) with or without adsorbed O species. $N_O$ is amount of adsorbed O species. $\mu_{O_2}$ is chemical potential of O₂, which can be expressed by following Eqs. (10) and (11):

$$\mu_{O_2}(T,p) = E_{O_2}^{DFT} + E_{O_2}^{ZPE} + \Delta\mu_{O_2}(T,p^0) + k_B T \ln\left(\frac{p_{O_2}}{p^0}\right) \qquad (10)$$

$$\Delta\mu_{O_2}(T,p^0) = [H_{O_2}(T,p^0) - H_{O_2}(OK,p^0)] - T[S_{O_2}(T,p^0) - S_{O_2}(OK,p^0)] \qquad (11)$$

where $E_{O_2}^{DFT}$ and $E_{O_2}^{ZPE}$ are the DFT energy and zero-point energy of O₂. $H_{O_2}$ and $S_{O_2}$ represent the enthalpy and entropy of gaseous O₂ at the respective temperature and pressure, which can be obtained from the NIST database (NIST-JANAF Thermochemical Tables, NIST Standard Reference Database 13, Last Update to Data Content: 1998, DOI: 10.18434/T42S31. Website: https://janaf.nist.gov/). $T = 1073K$, $p^0 = 1atm$. $k_B$ is the Boltzmann constant.

Here, $E_{O_2}^{ZPE}$ was calculated based on the vibrational frequency using the following Eq. (12):

$$E_{O_2}^{ZPE} = \sum_{i=1}^{k} \frac{\hbar\upsilon_i}{2} \qquad (12)$$

where $\upsilon_i$ is the vibrational frequency of O₂. $\hbar$ is Planck constant.

The dissociative adsorption energy of O₂ ($E_{O_2}$) was calculated as

$$E_{O_2} = E_{*2O} - \left(E_* + E_{O_2}\right) \qquad (13)$$

where $E_{*2O}$ is the energy of 2 O atoms absorbed on the surface. $E_{O_2}$ is the energy of gas phase O₂. $E_*$ is the energy of the clean surface.

Additionally, the surfaces obtained from phase diagram underwent three repeated AIMD simulations with NVT ensemble at 1073 K to assess the stability of surfaces containing adsorbed oxygen. To expedite the MD process, a 1 × 1 × 1 Gamma-centred k-points grid mesh was utilized.

## Data availability

The data supporting the findings of the study are available within the paper and its Supplementary Information. Source data are provided with this paper.

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

## Acknowledgements

This work is supported by the National Key R&D Program of China (2021YFA1501302), the National Science Foundation of China (No. 22121004, 22122808, and U1862207), the Haihe Laboratory of Sustainable Chemical Transformations, the Program of Introducing Talents of Discipline to Universities (BP0618007) and the XPLORER PRIZE. The authors thank the assistance from 1W1B beamline staff of Beijing Synchrotron Radiation Facility (BSRF) for the XAFS spectra test. Thanks for Mr. Kui Lin at Analysis and Testing Center of Tianjin University for quasi in-situ XPS test and thank Dr. Guodong Sun, Dr. Wei Wang and Dr. Xianhui Wang for fruitful discussion.

## Author contributions

J.G. conceived and coordinated the research. Z.D. contributed to catalyst synthesis, catalytic experiments and all the characterizations and analyzed the data. C.P., Z.S. and X.Z. participated in the discussion of the catalytic behavior. Z.D. and S.C. contributed to the kinetic tests. T.Y. and Z.-J.Z. carried out DFT calculations. Z.D., S.C., D.F., Z.-J.Z. and J.G. wrote the manuscript. All authors participated in the discussion of the research.

## Competing interests

The authors declare no competing interests.
