## [Peer Review File · Nature Communications]

Atomically Dispersed MoNi Alloy Catalyst for Partial Oxidation of MethaneREVIEWER COMMENTS

Reviewer #1 (Remarks to the Author):

The manuscript reports the synthesis, characterisation and testing of an atomically dispersed MoNi alloy with the results demonstrating that both selectivity and yield are high for the dispersed catalyst system.

The characterisation of the catalyst appears appropriate spanning a range of techniques including spectroscopy and in-situ kinetics. All of this work appears to confirm that the catalyst is an atomically dispersed alloy and that this alloy provides a superior performance.

However, conversion-selectivity comparisons appear at only one which limits the performance evaluation. The pure Ni catalyst provided 97% conversion and 90% selectivity vs 95% and 97% respectively for the alloy. While the improvement in performance is welcome it would be interesting to evaluate the impact of system conditions, in particular temperature, on these two cases particularly as the latter catalyst slowly deactivates over time and appears after 20h to be similar to the single metal in terms of conversion. Some supplementary data (e.g. figure S5) suggests changes to the catalyst over time. While others e.g. figure s9 show little change, note figure S9 refers to the Si K but the images show the O K.

While it is stated that the performance of the catalyst is superior in terms of selectivity when compared to previous reports, table S3 does show that some catalyst systems do provide both higher selectivity and conversion albeit under different conditions, loadings and when ignoring other factors such as deactivation.

Overall the work is good and does appear to show the creation and superior performance of the alloy. The characterisation and evaluation of the catalyst is good although some additional comparisons between the Ni/SBA-15 and MoNi alloy over different temperatures would help to demonstrate how much better or not the alloy is. At present the differences appear relatively small and the significance in terms of the overall impact to POM catalyst design is limited.

The quality of the English is relatively poor throughout making it more complicated to review. The manuscript could use a thorough proof reading to correct this prior to any resubmission.

Reviewer #2 (Remarks to the Author):

Ding et al. report an atomically dispersed MoNi alloy catalyst for the partial oxidation of methane, which allows the regulation of O* occupation, leading to high catalytic performance. The catalyst was studied by a combination of ex-situ/in-situ characterizations, and computational studies to rationalize the high catalytic performance. Contents in the theoretical section is quite route, and it also lacks more detailed analysis to provide better understanding, so significant additional works and corresponding revisions are necessary.

- (1) There should be more discussion on comparing the performance of the MoNi alloy catalyst developed in this work with other Ni-based catalysts for this reaction.
- (2) In Figure S14, why are the formation energies of the various 2Mosub structures have such a large spread? The authors presented the data, but provided little explanations or understanding. How much energy difference is calculated between the 1Mosurf and 1Mosub with a single Mo dopant?
- (3) On line 240-241, the authors found "The average adsorption energy (Eads) towards O* for MoNi alloy of -1.30 eV is higher than Eads for Ni of -1.41 eV". To my understanding, the O adsorption energy on the MoNi alloy was calculated for the MoNi alloy surface with pre-adsorbed O atoms, so are

there any pre-adsorbed O atoms on the Ni surface? Similar study on the phase diagram of Ni under reaction condition should be performed. Furthermore, is the MoNi alloy surface with pre-adsorbed O atoms stable under the relative condition? This is may be shown by ab initio molecular dynamics calculations? This is important, as pre-adsorbed O atoms cannot be directly observed in the experiment. Furthermore, how does the O adsorption changes as O atoms are added to the MoNi alloy surface?

(4) On line 243-245, the authors stated "Eads for CH₃* species adsorbed on the Ni-Ni bridge site of Ni and O site of MoNi alloy correspond to value of 1.75 and 1.49 eV". From this, they concluded "CH₄ activation ability was rarely weakened for MoNi alloy", which is actually confusing to me. Why "rarely weakened"? Are there situations when they are actually weakened? More importantly, more evidence than Eads for CH₃* species needs to be provided to judge CH₄ activation ability, for example, what is difference between the energy barrier for CH₄ dissociation or other step if CH₄ dissociation is not the rate-determining step? The calculations of the transition states for some key steps are at least necessary.

(5) On the Computational details section, a force convergence of 0.05 eV/Ang is too high, and 0.02 eV/Ang should be used. How was the chemical potential of O₂ actually calculated? That is, how did the authors calculate the H and S values in Eq. 10.

Reviewer 1:

General Comments R1: *The manuscript reports the synthesis, characterisation and testing of an atomically dispersed MoNi alloy with the results demonstrating that both selectivity and yield are high for the dispersed catalyst system. The characterisation of the catalyst appears appropriate spanning a range of techniques including spectroscopy and in-situ kinetics. All of this work appears to confirm that the catalyst is an atomically dispersed alloy and that this alloy provides a superior performance.*

Response: We thank the reviewer for the positive evaluations.

Specific comments R1-1: *The quality of the English is relatively poor throughout making it more complicated to review. The manuscript could use a thorough proof reading to correct this prior to any resubmission.*

Response: We thank the reviewer for this comment. We have carefully revised the manuscript accordingly. All revised words are highlighted in the manuscript.

Specific Comment R1-2: *However, conversion-selectivity comparisons appear at only one which limits the performance evaluation. The pure Ni catalyst provided 97% conversion and 90% selectivity vs 95% and 97% respectively for the alloy. While the improvement in performance is welcome it would be interesting to evaluate the impact of system conditions, in particular temperature, on these two cases particularly as the latter catalyst slowly deactivates over time and appears after 20h to be similar to the single metal in terms of conversion.*

Response: We thank the reviewer for valuable suggestions. To show the comprehensive comparisons of conversion-selectivity over pure Ni and the MoNi alloy catalysts, we have evaluated the impact of system conditions encompassing temperature and gaseous hourly space velocity (GHSV), as shown in the revised Supplementary Fig. S9-S10.

It is noted that the comparison emphasizes the superior performance and stability of AD MoNi alloy/SBA-15 over Ni/SBA-15 under the investigated conditions. It appears that the AD MoNi alloy/SBA-15 maintains stability in the temperature range from 800 to 700 °C under various GHSV ranging from 15,000 to 48,000 mL_{CH₄} g_{cat}⁻¹ h⁻¹. In contrast, the Ni/SBA-15 catalyst exhibits a slight deactivation for methane conversion from 74.4 % to 73.1 % after a 2 h reaction starting at 700 °C with a GHSV of 15,000 mL_{CH₄} g_{cat}⁻¹ h⁻¹. As the GHSV increases to 24,000 mL_{CH₄} g_{cat}⁻¹ h⁻¹ at 800 °C, Ni/SBA-15 shows noticeable deactivation, reducing methane conversion from 85.5% to 80.3% after a 2 h reaction, lower than that of 92.5% for AD MoNi alloy/SBA-15, as illustrated in revised Figure S9E. Additional comparison tests are provided in the revised Figure S10, and detailed data can be found in Tables S5-S7. These results suggest that the AD MoNi alloy/SBA-15 exhibits superior stability and performance compared to Ni/SBA-15 under the investigated reaction conditions. We add discussion in the revised manuscript as follows:

Figure S9 | Performance comparison between AD Mo/Ni alloy/SBA15 and Ni/SBA15. On-stream profiles of (A) Conversion and (B) Selectivity, and comparison of (C) Conversion and (D) Selectivity with GHSV of 15,000 mL_{CH₄} g_{cat}⁻¹ h⁻¹. On-stream profiles of (E) Conversion and (F) Selectivity, and comparison of (G) Conversion and (H) Selectivity with GHSV of 24,000 mL_{CH₄} g_{cat}⁻¹ h⁻¹ for AD Mo/Ni alloy/SBA15 and Ni/SBA15, respectively.

Figure S10 | Performance comparison between AD Mo/Ni alloy/SBA15 and Ni/SBA15. On-stream profiles of (A) Conversion and (B) Selectivity, and comparison of (C) Conversion and (D) Selectivity with GHSV of 48,000 mL_{CH₄} g_{cat}⁻¹ h⁻¹. On-stream profiles of (E) Conversion and (F) Selectivity of AD Mo/Ni alloy/SBA15 with temperature switched from 850 to 700 °C and GHSV from 24,000 to 48,000 mL_{CH₄} g_{cat}⁻¹ h⁻¹.

Table S5 Performance comparison between AD Mo/Ni alloy/SBA15 and Ni/SBA15 at different temperature with GHSV of 15,000 mL_{CH₄} g_{cat}⁻¹ h⁻¹.

Temperature	Initial CH ₄ conversion (%)		Initial H ₂ selectivity (%)		Initial CO selectivity (%)	
	AD Mo/Ni alloy/SBA15	Ni/SBA15	AD Mo/Ni alloy/SBA15	Ni/SBA15	AD Mo/Ni alloy/SBA15	Ni/SBA15
800 °C	93.9	95.9	93.2	91.0	96.3	92.5
750 °C	90.5	88.1	91.2	89.3	90.9	87.8
700 °C	80.9	74.4	88.4	85.2	84.1	77.7

Table S6 Performance comparison between AD Mo/Ni alloy/SBA15 and Ni/SBA15 at different temperature with GHSV of 24,000 mL_{CH4} g_{cat}⁻¹ h⁻¹

Temperature	Initial CH ₄ conversion (%)		Initial H ₂ selectivity (%)		Initial CO selectivity (%)	
	AD Mo/Ni alloy/SBA15	Ni/SBA15	AD Mo/Ni alloy/SBA15	Ni/SBA15	AD Mo/Ni alloy/SBA15	Ni/SBA15
800 °C	92.5	85.5	95.3	87.8	92.1	88.7
750 °C	83.9	68.5	88.2	78.2	84.5	84.3
700 °C	68.3	52.7	80.5	63.5	71.8	59.7

Table S7 Performance comparison between AD Mo/Ni alloy/SBA15 and Ni/SBA15 at different temperature with GHSV of 48,000 mL_{CH4} g_{cat}⁻¹ h⁻¹

Temperature	Initial CH ₄ conversion (%)		Initial H ₂ selectivity (%)		Initial CO selectivity (%)	
	AD Mo/Ni alloy/SBA15	Ni/SBA15	AD Mo/Ni alloy/SBA15	Ni/SBA15	AD Mo/Ni alloy/SBA15	Ni/SBA15
800 °C	88.9	84.3	88.6	87.4	89.0	88.7
750 °C	82.0	72.7	85.8	82.1	84.8	80.8
700 °C	73.4	59.6	81.5	74.4	77.8	69.2

Line 12-30, Page 7:

The catalytic performance comparison is detailed in Table S3 and S4, highlighting the superior selectivity of the AD MoNi alloy/SBA-15 catalyst compared to previous reports. This catalyst demonstrates a remarkable selectivity of over 97%, higher than the most reported Ni-based catalysts and comparable to noble metal-based catalysts.

The catalytic performance under varying conditions, encompassing temperature and gaseous hourly space velocity (GHSV), for Ni/SBA-15 and AD MoNi alloy/SBA-15, are presented in Supplementary Fig. 9 and Supplementary Fig. 10. It appears that the AD MoNi alloy/SBA-15 maintains stability in the temperature range from 800 to 700 °C and under various GHSV ranging from 15,000 to 48,000 mL_{CH4} g_{cat}⁻¹ h⁻¹. In contrast, the Ni/SBA-15 catalyst experiences a slight deactivation for methane conversion from 74.4% to 73.1 % after 2 h reaction starting at 700 °C with a GHSV of 15,000 mL_{CH4} g_{cat}⁻¹ h⁻¹. As the GHSV increases to 24,000 mL_{CH4} g_{cat}⁻¹ h⁻¹ at the same temperature (800 °C), Ni/SBA-15 shows noticeable deactivation, reducing methane conversion from 85.5% to 80.3% after a 2 h reaction (Supplementary Fig. 9E), lower than that of 92.5% for AD MoNi alloy/SBA-15. Therefore, the comparison emphasizes the superior performance and stability of AD MoNi alloy/SBA-15 over Ni/SBA-15 under the investigated conditions shown in Table S5-S7.

Specific Comment R1-3: Some supplementary data (e.g. figure S5) suggests changes to the catalyst over time. While others e.g. figure S9 show little change, note figure S9 refers to the Si K but the images show the O K.

Response: We appreciate the reviewer's insightful comments.

- 1) It is suggested that XPS spectra in Figure S5 primarily reflect the formation of Mo oxide on the catalysts surface for the spent catalyst. Considering the absence of Mo oxide peaks in XRD

patterns as shown in revised Figure S11 (previous Figure S9), it is believed that some of the surface Mo in the alloy could convert into amorphous Mo oxides after the reaction as shown by the XPS spectra (Figure S5). Therefore, despite the formation of Mo oxide evidenced by XPS, there are no diffraction peaks assigned to Mo oxide. For a more comprehensive understanding of structural evolution, XRD patterns for fresh, spent, and deactivated species for AD MoNi/SBA-15 and Ni/SBA-15 are provided. It is noted that Ni/SBA-15 catalyst underwent oxidation, transforming into the NiO phase after the reaction. In contrast, the AD MoNi alloy/SBA-15 maintained its alloy phase, displaying a left shift of 0.3° located at 44.2° compared to the metallic Ni phase at 44.5° .

Figure S11 (D) XRD for deactivated, spent (20h test) and fresh Ni/SBA-15 catalysts, and the deactivated, spent (20h test) and fresh AD MoNi alloy/SBA-15 catalysts.

2) As for previous Figure S9 caption, we apologized for the misleading. We have corrected them in revised Figure S11 caption as following:

Figure S11 | Elemental distribution for fresh and spent catalyst. EDS mapping for (A) Ni K, (B) O K, (C) merge of pure Ni/SBA-15 catalysts (D) XRD for deactivated, spent (20 h test) and fresh Ni/SBA-15 catalysts, and the deactivated, spent (20 h test) and fresh AD MoNi alloy/SBA-15 catalysts. EDS mapping of (E) Ni K, (F) Mo L, (G) O K and (H) merge of AD MoNi alloy/SBA-15. EDS mapping of (I) Ni K, (J) Mo L, (K) O K and (L) merge of spent AD MoNi alloy/SBA-15 for POM reaction.

Specific Comment R1-4: While it is stated that the performance of the catalyst is superior in terms of selectivity when compared to previous reports, table S3 does show that some catalyst systems do provide both higher selectivity and conversion albeit under different conditions, loadings and when ignoring other factors such as deactivation.

Response: We thank the reviewer for these suggestions. We apologized for the unjustified statement for the designed MoNi catalysts and previous reported catalysts. As shown in table S3, the selectivity of MoNi catalysts is superior to the most reported catalysts under different conditions. We have restated this in the revised manuscript.

Line 14-16, Page 7:

This catalyst demonstrates a remarkable selectivity of over 97%, higher than the most reported Ni-based catalysts and comparable to noble metal-based catalysts.

Specific Comment R1-5: *Overall the work is good and does appear to show the creation and superior performance of the alloy. The characterisation and evaluation of the catalyst is good although some additional comparisons between the Ni/SBA-15 and MoNi alloy over different temperatures would help to demonstrate how much better or not the alloy is. At present the differences appear relatively small and the significance in terms of the overall impact to POM catalyst design is limited.*

Response: We thank the reviewer for this positive evaluation. We have supplemented the comparisons of Ni/SBA-15 and MoNi alloy by evaluating the impact of system conditions, encompassing temperature and gaseous hourly space velocity (GHSV) (Also see in comment R1-1 and response). Under the evaluated temperatures (700-800 °C) and GHSV (15,000-48,000 mL_{CH₄} g_{cat}⁻¹ h⁻¹), the MoNi alloy catalysts exhibited superior catalytic performance than Ni/SBA-15.

Reviewer 1:

General Comments R2: *Ding et al. report an atomically dispersed MoNi alloy catalyst for the partial oxidation of methane, which allows the regulation of O* occupation, leading to high catalytic performance. The catalyst was studied by a combination of ex-situ/in-situ characterizations, and computational studies to rationalize the high catalytic performance. Contents in the theoretical section is quite route, and it also lacks more detailed analysis to provide better understanding, so significant additional works and corresponding revisions are necessary.*

Response: We thank the reviewer for this positive evaluation. We have carefully addressed the comments raised by the reviewer and revised the manuscript accordingly.

Specific Comment R2-1: *There should be more discussion on comparing the performance of the MoNi alloy catalyst developed in this work with other Ni-based catalysts for this reaction.*

Response: We thank the reviewer for this suggestion. As showed in specific comment R1-1 and response, we supplemented performance comparison under varying conditions towards AD MoNi/SBA-15 and Ni/SBA-15 listed in revised Figure S9-S10 and Table S5-S7 to fairly evaluate the performance of MoNi alloy with Ni-based catalyst. This MoNi/SBA-15 catalyst demonstrates a remarkable selectivity of over 97%, higher than the Ni Ni/SBA-15 catalysts. We also added the discussion on comparing the performance with other Ni-based catalysts as follow:

Line 12-30, Page 7:

The catalytic performance comparison is detailed in Table S3 and S4, highlighting the superior selectivity of the AD MoNi alloy/SBA-15 catalyst compared to previous reports. This catalyst demonstrates a remarkable selectivity of over 97%, higher than the most reported Ni-based catalysts and comparable to noble metal-based catalysts.

The catalytic performance under varying conditions, encompassing temperature and gaseous hourly space velocity (GHSV), for Ni/SBA-15 and AD MoNi alloy/SBA-15, are presented in Supplementary Fig. 9 and Supplementary Fig. 10. It appears that the AD MoNi alloy/SBA-15 maintains stability in the temperature range from 800 to 700 °C and under various GHSV ranging from 15,000 to 48,000 mL_{CH₄} g_{cat}⁻¹ h⁻¹. In contrast, the Ni/SBA-15 catalyst experiences a slight deactivation for methane conversion from 74.4% to 73.1 % after 2 h reaction starting at 700 °C with a GHSV of 15,000 mL_{CH₄} g_{cat}⁻¹ h⁻¹. As the GHSV increases to 24,000 mL_{CH₄} g_{cat}⁻¹ h⁻¹ at the same temperature (800 °C), Ni/SBA-15 shows noticeable deactivation, reducing methane conversion from 85.5% to 80.3% after a 2 h reaction (Supplementary Fig. 9E), lower than that of 93.9% for AD MoNi alloy/SBA-15. Therefore, the comparison emphasizes the superior performance and stability of AD MoNi alloy/SBA-15 over Ni/SBA-15 under the investigated conditions shown in Table S5-S7.

Specific Comment R2-2: In Figure S14, why are the formation energies of the various 2Mosub structures have such a large spread? The authors presented the data, but provided little explanations or understanding. How much energy difference is calculated between the 1Mosurf and 1Mosub with a single Mo dopant?

Response: We appreciate the reviewer's valuable comments. In previous Figure S14 (revised Figure S16), the structure with the highest energy corresponds to one Mo atom in the surface layer and another Mo atom in the fifth layer. The elevated energy in this configuration is attributed to the fact that during the calculations, we kept the bottom two layers atoms, those are fourth and fifth layer, fixed to simulate the bulk structure, leading to a lack of local structure optimization for

the Mo atom in the fifth layer, which exhibits the highest energy. Comparatively, the lowest energy structure occurs when both Mo atoms are positioned in the subsurface layer. The energy difference between these two configurations is 1.81 eV. Interestingly, when one Mo atom is at the surface and the other in a deeper layer (excluding the fifth layer), 1Mo_{surf}-1Mo_{sub} models consistently shows lower energy compared to 2Mo_{surf} models. Notably, the energy difference between 1Mo_{surf} and 1Mo_{sub} is 0.72 eV, whereas the energy difference between 2Mo_{surf} and 2Mo_{sub} is 1.50 eV, approximately twice the value of 0.72 eV. This observation underscores the reliability of our calculations. We have added the description and updated in the revised supporting information (Figure S16) shown as:

Figure S16. Formation energy for possible structure determination. (A) Energies of surface models with 2 Mo atoms at different position. Inset represent corresponding schematic diagram (The structure marked by purple circle represents the most stable structure). Possible structure for AD MoNi alloy and its energy, where the two Mo atoms located at sub-surface for MoNi structure was the most stable structure with most negative energy. (B) Energy difference of -0.72 eV between 1Mo_{sub} and 1Mo_{surf} with a single Mo atom.

Specific Comment R2-3: On line 240-241, the authors found “The average adsorption energy (E_{ads}) towards O* for MoNi alloy of -1.30 eV is higher than E_{ads} for Ni of -1.41 eV”. To my understanding, the O adsorption energy on the MoNi alloy was calculated for the MoNi alloy surface with pre-adsorbed O atoms, so are there any pre-adsorbed O atoms on the Ni surface? Similar study on the phase diagram of Ni under reaction condition should be performed. Furthermore, is the MoNi alloy surface with pre-adsorbed O atoms stable under the relative condition? This is may be shown by ab initio molecular dynamics calculations? This is important, as pre-adsorbed O atoms cannot be directly observed in the experiment. Furthermore, how does the O adsorption changes as O atoms are added to the MoNi alloy surface?

Response: We appreciate the reviewer's valuable comments.

- 1) We supplemented the phase diagram of Ni (100) under reaction conditions and observed the absence of pre-adsorbed oxygen species on pure Ni surfaces as shown in the revised Figure S19. Therefore, choosing the O adsorption energy on pure Ni as a reference for comparison is appropriate.

Figure S19. Phase diagram for Ni. Phase diagram for possible structure with practical O₂ partial pressure of ~0.125 under reaction condition at 800 °C.

2) Furthermore, to illustrate the stability of oxygen adsorption on MoNi(100), we conducted three repeated 20 ps ab initio molecular dynamics (AIMD) simulations with NVT ensemble at 1073 K. As shown in the revised Figure S20, the MoNi (100) exhibits high stability, with nearly constant energy. The structure also maintains a similar configuration despite the presence of slight atomic movement.

Figure S20. Ab initio molecular dynamics (AIMD) for AD MoNi. (A-C) Three repeated AIMD simulations were conducted at 1073 K within the NVT ensemble for 20ps. (D-F) The atomic structures represent MoNi (100) after simulation. Pre-adsorbed O atoms remain stable for 20 ps under reaction condition.

3) To further elucidate the structural evolution process of the MoNi(100) surface, we compared the trends of average oxygen adsorption energy on the $1\text{Mo}_{\text{surf}}-1\text{Mo}_{\text{sub}}$ surfaces with varying oxygen coverage. We found that the $1\text{Mo}_{\text{surf}}-1\text{Mo}_{\text{sub}}$ surfaces remain stable only when the surface is covered with $2/9$ and $3/9$ of oxygen atoms (only 2-3 O atoms coverage is negative value) as supplemented in the revised Figures S17 and S18. This observation aligns with the predicted structures from the phase diagram of MoNi(100), as depicted in Figure 5E. It

indicates that in the absence of surface-adsorbed oxygen, Mo tends to reside in the subsurface layer. As surface oxygen coverage increases, Mo gradually migrates to the surface. Once the surface oxygen coverage exceeds 1/3, both Mo atoms segregate to the surface.

Figure S17. Segregation energy for possible structure determination. Segregation energy of MoNi (100) with different amount of adsorbed O species.

Figure S18. Gibbs free adsorption energy. The variations of average oxygen Gibbs free adsorption energy on 1Mosurf-1Mosub surface with different surface oxygen coverage.

Here, the average oxygen Gibbs free adsorption energy (G_{ads}) was calculated by following equation:

$$G_{\text{ads}} = [G_{*O} - (G_* + \frac{N_O}{2} \mu_{O_2})] / N_O$$

Where G_{*O} is the Gibbs free energies of 1Mo_{surf}-1Mo_{sub} surfaces with N_O adsorbed O. G_* is the Gibbs free energies of 2Mo_{sub} surfaces without adsorbed O. N_O is the number of adsorbed O. μ_{O_2} is chemical potential of O₂, which can be calculated according to method section computational details parts outlined in revised manuscript.

Finally, we further validated all phase diagram data and refined certain statements. The updated description has been included in the revised manuscript with highlight as following:

Line 10-23, Page 11, Line 1-8, Page 12:

In addition, a density functional theory (DFT) study was conducted to provide a molecular-

level insight into the mechanism of the partial oxidation of methane (POM) process. Energy calculations for a series of slab models were performed to determine the position of Mo in the Ni lattice. It was observed that Mo atoms located at the subsurface exhibited the most negative energy, as depicted in Supplementary Fig. 16. This suggests a preference for Mo atoms to reside within the subsurface layer of the MoNi (100). However, under reaction conditions, the presence of surface-adsorbed oxygen species may induce the segregation of Mo atoms from the subsurface to the surface. Calculations of the segregation energy (E_{seg}) and oxygen Gibbs free adsorption energy (E_{O-ads}) revealed that as the number of surface O* species increased, the E_{seg} of Mo became more negative and E_{O-ads} is negative for 2-3 O atoms (Supplementary Fig. 17 and Supplementary Fig. 18). This indicates that Mo is more likely to migrate to the surface and occupy surface positions and only when 2-3 O atoms coverage could MoNi (100) remain stable. Subsequently, phase diagrams for MoNi (100) and Ni (100) were computed to elucidate the authentic surface structures of the catalysts under reaction conditions (Fig. 5E and Supplementary Fig. 19). Specifically, it was observed that there are no adsorbed oxygen species on the Ni (100) surface, while two adsorbed oxygen species are present on the MoNi (100) surface. To confirm the stability of the MoNi structure with two pre-adsorbed O species, three repeated ab initio molecular dynamics (AIMD) simulations were conducted (Supplementary Fig. 20). The results of these simulations support the stability of the MoNi structure under the specified conditions.

Fig. 5 | Kinetic analysis and DFT calculation. (A) CH₄-TPSR for pure Ni/SBA-15 and (B) AD MoNi alloy/SBA-15 catalyst. (C) Reaction order for CH₄ and (D) O₂. (E) Phase diagram for AD MoNi alloy with practical O₂ partial pressure of ~0.125 under reaction condition at 800 °C. (F) possible structure of phase diagram in (E). Red square in (E) and (F) represents real structure according to calculated phase diagram. Ni: blue ball; Mo: bright red ball; O: deep red ball; C: black; H: white ball.

Specific Comment R2-4: On line 243-245, the authors stated “E_{ads} for CH₃* species adsorbed on the Ni-Ni bridge site of Ni and O site of MoNi alloy correspond to value of 1.75 and 1.49 eV”. From this, they concluded “CH₄ activation ability was rarely weakened for MoNi alloy”, which is actually confusing to me. Why “rarely weakened”? Are there situations when they are actually weakened? More importantly, more evidence than E_{ads} for CH₃* species needs to be provided to judge CH₄ activation ability, for example, what is difference between the energy barrier for CH₄

dissociation or other step if CH₄ dissociation is not the rate-determining step? The calculations of the transition states for some key steps are at least necessary.

Response: We thank the reviewer for this insightful comment.

1) To elucidate the role of Mo doping, we supplemented the reaction pathways and transition states of methane dehydrogenation and further reaction on MoNi (100) and Ni (100), as depicted in revised Figure S21 and Figure S22. It can be observed that the activation energy barriers for methane on MoNi (100) and Ni(100) are 1.12 and 1.09 eV, respectively, indicating comparable methane activation capabilities on these two surfaces, albeit slightly reduced on MoNi (100). For the methane combustion reaction, by comparing the rate-determining steps (RDS) on both surfaces involving the reaction of H* with O* to form HO*, we found that the energy barriers for RDS are very similar, measuring 1.17 eV and 1.21 eV, respectively. However, in the subsequent methane reforming reaction, methane dehydrogenation behavior resembles that of methane combustion, but the rate-controlling step for the entire reaction becomes the elementary step of C* + O* → CO*, which is consistent with the literature findings (*Ind. Eng. Chem. Res.* 2015, 54, 5901-5913, *ACS Catal.* 2016, 6, 10, 6730-6738). Therefore, we deduce that the introduction of Mo has minimal influence on the consecutive dehydrogenation process of methane, indicating very similar methane activation ability on MoNi (100) and Ni(100). This means CH₄ activation ability was rarely weakened for MoNi alloy than Ni. However, the introduction of Mo reduces the energy barrier for CO* formation, at 1.50 eV for MoNi and 1.76 eV for Ni, thereby favoring enhanced selectivity (revised Figure S23A-B).

Figure S21 | Reaction pathways for MoNi (100) and Ni (100). Transition state of methane and oxygen on (A) MoNi (100) and (B) Ni (100).

Figure S22 | Reaction pathways for MoNi (100) and Ni (100). Schematic structures for transition state of methane and oxygen on (A) MoNi (100) and (B) Ni (100).

- 2) Through further calculation of O_2 adsorption energies (revised Figure S23C), we found that O_2 directly dissociates and adsorbs on both two surfaces, with the dissociation energy significantly lower on pure MoNi (100) compared to pure Ni(100), at -3.12 and -1.45 eV, respectively. This indicates that the presence of Mo facilitates the dissociation of O_2 into two O^* species adsorbed on Mo, thereby promoting their participation in H_2O formation. Additionally, upon consumption of these two O^* species on MoNi (100), they can be rapidly replenished by O_2 .

Figure S23. (A) Reaction pathways of C^* with O^* during methane reforming on (A) MoNi and (B) Ni. (C) The dissociative adsorption energy of O_2 on pure MoNi, pure Ni and MoNi-20.

Hence, the introduction of Mo rarely weakens the ability for CH_4 activation but serves two main purposes: firstly, aiding in the activation of O_2 , effectively releasing active sites on Ni surfaces for CH_4 dehydrogenation and alleviating competition between CH_4 and O_2 for Ni surface; secondly, promoting CO formation, thus enhancing selectivity. To further explore the effects of Mo doping, we investigated O_2 adsorption behavior on MoNi (100) and Ni (100). Our findings reveal direct dissociation of O_2 on both surfaces, with significantly lower dissociation energy on

pure MoNi compared to Ni and MoNi-2O surfaces (Supplementary Fig. 23C). This implies rapid replenishment of pre-adsorbed O* species on MoNi after consumption during the reaction, facilitating reaction progress. Thus, Mo primarily promotes the activation of O₂ effectively releasing active sites on Ni sites for CH₄ activation and favors the enhancement of syngas selectivity, consistent with experimental results.

We have added the description in the revised manuscript with highlight as following:

Line 9-34, Page 12:

We conducted a calculation of the reaction pathway for methane dehydrogenation on Ni (100) and MoNi (100). Schematic structures of all transition states are provided in Supplementary Fig. 21 and Supplementary Fig. 22. The activation energy barrier for methane on MoNi (100) is slightly higher than that on Ni (100), measuring 1.12 eV and 1.09 eV, respectively. This suggests a modest weakening of methane activation ability in the MoNi alloy, in line with experimental findings (Fig. 5). Subsequently, CH_x* species undergo dehydrogenation on Ni sites until forming C*, which preferentially reacts with O₂ to form CO₂. Then, 2H* forms H₂O by interacting with pre-adsorbed O* on the MoNi surface. For methane combustion, the rate-determining step for both MoNi (100) and Ni (100) is the migration of H* from Ni to O*, forming HO*, with similar energy barriers observed for both surfaces⁵⁸. However, the elementary step of C* combining with O* to form CO* is identified as the rate-determining step in the subsequent methane reforming reaction^{58,59}, as shown in Supplementary Fig. 23.

On MoNi (100), the adsorbed O* on Mo first migrates to Ni before forming adsorbed CO*, rather than directly combining with C to form CO*. Additionally, the energy barrier of CO* formation on MoNi (100) is observed to be lower than that on the Ni (100) surface, indicating that MoNi promotes the formation of CO, thereby enhancing product selectivity. To further explore the effects of Mo doping, we investigated O₂ adsorption behavior on MoNi (100) and Ni (100). Our findings reveal direct dissociation of O₂ on both surfaces, with significantly lower dissociation energy on pure MoNi compared to Ni and MoNi-2O surfaces (Supplementary Fig. 23C). This implies rapid replenishment of pre-adsorbed O* species on MoNi after consumption during the reaction, facilitating reaction progress. Thus, Mo primarily promotes the activation of O₂, inhibits the dissociative adsorption of O₂ on Ni sites on MoNi, and favors the enhancement of syngas selectivity, consistent with experimental results.

Specific Comment R2-5: On the Computational details section, a force convergence of 0.05 eV/Ang is too high, and 0.02 eV/Ang should be used. How was the chemical potential of O₂ actually calculated? That is, how did the authors calculate the H and S values in Eq. 10.

Response: We express gratitude to the reviewer for providing this comment. We have reviewed all the input files, and it is noted that a force convergence criterion of 0.02 eV/A was employed for all calculations within the manuscript.

According to Eq. 10, we can determine the chemical potential of O₂ at reaction temperature and pressure. Here, $E_{O_2}^{DFT}$ represents the DFT energy of O₂ molecules, while $E_{O_2}^{ZPE}$ can be obtained using Eq. 12 based on the vibrational frequency of O₂. The enthalpy (H_{O_2}) and entropy (S_{O_2}) of O₂ were obtained from the NIST thermodynamic database at the reaction temperature and standard pressure. Then, substituting these values into Eq. 11 allows us to calculate $\Delta\mu_{O_2}(T, p^0)$. Finally, by substituting the partial pressure of oxygen under reaction conditions into $k_B T \ln(\frac{p_{O_2}}{p^0})$, we can determine the chemical potential of O₂ under reaction conditions. We have added the description

in method section computational details parts in the revised manuscript with highlight as following:

Line 20-22, Page 12:

We added Ref. 58 and 59.

Line 34, Page 18 to Line 1-8, Page 19:

We added Ref. 62-66. Line 3, Page 19:

Energies and forces for geometric optimization were converged to within 10^{-5} eV and 0.02 eV/Å, respectively.

Line 24-32, Page 19 to Line 1-3, Page 20:

$$\mu_{O_2}(T, p) = E_{O_2}^{DFT} + E_{O_2}^{ZPE} + \Delta\mu_{O_2}(T, p^0) + k_B T \ln\left(\frac{p_{O_2}}{p^0}\right) \quad (10)$$

$$\Delta\mu_{O_2}(T, p^0) = [H_{O_2}(T, p^0) - H_{O_2}(OK, p^0)] - T[S_{O_2}(T, p^0) - S_{O_2}(OK, p^0)] \quad (11)$$

where $E_{O_2}^{DFT}$ and $E_{O_2}^{ZPE}$ are the DFT energy and zero-point energy of O_2 . H_{O_2} and S_{O_2} represent the enthalpy and entropy of gaseous O_2 at the respective temperature and pressure, which can be obtained from the NIST database (NIST-JANAF Thermochemical Tables, NIST Standard Reference Database 13, Last Update to Data Content: 1998, DOI: 10.18434/T42S31. Website: <https://janaf.nist.gov/>). $T = 1073K$, $p^0 = 1 atm$. k_B is the Boltzmann constant. Here, $E_{O_2}^{ZPE}$ was calculated based on the vibrational frequency using the following equation:

$$E_{O_2}^{ZPE} = \sum_{i=1}^k \frac{\hbar v_i}{2} \quad (12)$$

where v_i is the vibrational frequency. \hbar is Planck constant.

Line 28-30, Page 19:

NIST-JANAF Thermochemical Tables, NIST Standard Reference Database 13, Last Update to Data Content: 1998, DOI: 10.18434/T42S31, <https://janaf.nist.gov/>

REVIEWERS' COMMENTS

Reviewer #2 (Remarks to the Author):

The authors have significantly revised their manuscript, and the additional calculations and discussion substantially improved the mechanistic understanding of their developed catalysts. I recommend its acceptance for publication.